# Prostate Cancer and Sleep Disorders: A Systematic Review

**DOI:** 10.3390/cancers14071784

**Published:** 2022-03-31

**Authors:** Davide Sparasci, Ilenia Napoli, Lorenzo Rossi, Ricardo Pereira-Mestre, Mauro Manconi, Giorgio Treglia, Laura Marandino, Margaret Ottaviano, Fabio Turco, Dylan Mangan, Silke Gillessen, Ursula Maria Vogl

**Affiliations:** 1Sleep Medicine Unit, Neurocenter of Southern Switzerland, Ente Ospedaliero Cantonale (EOC), 6900 Lugano, Switzerland; davide.sparasci@eoc.ch (D.S.); mauro.manconi@eoc.ch (M.M.); 2Oncology Institute of Southern Switzerland (IOSI), Ente Ospedaliero Cantonale (EOC), 6500 Bellinzona, Switzerland; ilenia-napoli@live.it (I.N.); lorenzo.rossi@eoc.ch (L.R.); ricardo.pereiramestre@eoc.ch (R.P.-M.); marandino.laura@hsr.it (L.M.); margaretottaviano@gmail.com (M.O.); fabio.turco@eoc.ch (F.T.); d.man@manchester.ac.uk (D.M.); silke.gillessen@eoc.ch (S.G.); 3Radiation Oncology Unit, Department of Biomedical, Dental Science, Morphological and Functional Imaging, University Hospital Messina, 98122 Messina, Italy; 4Institute of Oncology Research (IOR), 6500 Bellinzona, Switzerland; 5Faculty of Biomedical Sciences, Università della Svizzera Italiana, 6900 Lugano, Switzerland; giorgio.treglia@eoc.ch; 6Department of Neurology, University Hospital Inselspital, 3010 Bern, Switzerland; 7Academic Education, Research and Innovation Area, General Directorate, Ente Ospedaliero Cantonale (EOC), 6500 Bellinzona, Switzerland; 8Faculty of Biology and Medicine, University of Lausanne, 1005 Lausanne, Switzerland; 9Department of Medical Oncology, IRCCS San Raffaele Hospital, 20132 Milan, Italy; 10Department of Clinical Medicine and Surgery, University Federico II of Naples, 80138 Naples, Italy; 11Department of Oncology, Division of Medical Oncology, University of Turin San Luigi Gonzaga Hospital, Regione Gonzole, 10043 Orbassano, Italy; 12Division of Population Health, University of Manchester, Manchester M13 9PL, UK

**Keywords:** systematic review, sleep disorder, prostate cancer, androgen deprivation therapy, radiotherapy

## Abstract

**Simple Summary:**

Longer survival times for prostate cancer patients due to efficient treatments consisting of local radiotherapy, prostatectomy and androgen-deprivation therapy, as well as androgen-receptor-targeted agents, increases the importance of side effect management. Sleep disturbances are higher in this group than the general population and no clear mechanism(s) explains this. This systematic review finds a reported effect in 14 of 16 included studies on sleep quality changes for these patients. All reported treatments showed some kind of negative effect on sleep quality, including ADT. Limitations are discussed and recommendations made for progressing the understanding and then for mitigation strategies of these side effects.

**Abstract:**

Prostate cancer (PCa) treatment involves multiple strategies depending on the disease’s stage. Androgen deprivation therapy (ADT) remains the gold standard for advanced and metastatic stages. Sleep quality has been suggested as being additionally influenced also by local radiotherapy, prostatectomy and androgen-receptor (AR)-targeted agents. We performed a systematic review exploring the landscape of studies published between 1 January 1990 and 31 July 2021, investigating sleep disturbances in PCa patients receiving active treatments, including the influence of hormonal therapy on sleep quality as a factor affecting their quality of life. Out of 45 articles identified, 16 studies were selected, which recruited patients with PCa, undergoing active treatment in either a prospective longitudinal or cross-sectional study. Development of sleep disorders or changes in sleep quality were reported in 14 out of 16 trials included. Only five trials included objective measurements such as actigraphy, mostly at one time point and without a baseline assessment. Limitations to be addressed are the small number of existing trials, lack of randomized trials and heterogeneity of methodologies used. This systematic review outlines the lack of prospective trials investigating sleep disorders, with a rigorous methodology, in homogeneous cohorts of PCa patients. Future trials are needed to clarify the prevalence and impact of this side effect of PCa treatments.

## 1. Introduction

Sleep is a circadian physiological process, vital to homeostasis and brain neuroplasticity. Chronic sleep deprivation has serious negative impacts on health, quality of life and neuro-cognitive performance [1,2,3,4,5]. Sleep disorders are more frequent in patients with cancer than in the general population, with respective prevalence rates estimated between 30% and 50% depending on the specific oncological diseases [6,7], and 10% and 15% in the general population [8].

Savard and colleagues [9] evaluated the prevalence and course of insomnia (symptoms and syndrome) in different types of cancer over a period of 18 months. The results showed that the prevalence of insomnia was higher throughout the duration of the study in patients with various cancer types (21% to 28%) than in the general population [10,11]. The highest rates of insomnia were shown in patients with breast cancer (42% to 69%) and gynecological cancer (33% to 68%), followed by prostate cancer (25% to 39%).

Prostate cancer (PCa) represents the second most common cancer in males, with approximately 1,276,106 newly diagnosed cases worldwide in 2018 [12]. In the European Union, the age-standardized mortality rate of prostate cancer is 10/100,000, declining by 7.1% since 2015 [13], most likely due to new treatment options and earlier diagnosis [14]. Therapeutic options are manifold, depending on the disease stage, and include prostatectomy, radiotherapy, hormonal therapy with luteinizing hormone-releasing hormone (LHRH) agonists or antagonists, as well as novel androgen receptor (AR)-targeting agents, chemotherapy and, more recently, also molecularly targeted agents and radiopharmaceuticals. For prostate cancer patients with locally advanced prostate tumors showing aggressive pathological features, radiotherapy is a valid treatment option, both for localized disease when used alone, or in the adjuvant setting following radical prostatectomy (RP). It is also conducted as a salvage modality in patients with biochemical recurrence after RP. There are several side effects caused by radiotherapy, generally due to irritation of the organs in proximity to the target. How radiation therapy causes sleep disturbances is not yet fully understood.

Androgen deprivation therapy (ADT) is the gold standard in patients with metastatic prostate cancer, an additional treatment option in men with biochemical relapse, following local radical treatment and used as neoadjuvant therapy for locally advanced disease in addition to radiotherapy [15]. Androgen deprivation has been a standard for treating prostate cancer since the Nobel-prize-winning discovery of the androgen-sensitive nature of the disease [16]. In high-risk localized disease and/or more advanced stages, combination treatments are utilized, adapted to the cancer stage, patient’s age and individual comorbidities. Due to the development of novel drugs and treatment lines in the metastatic stage, the median survival and therefore treatment duration of ADT has increased significantly [17,18,19,20,21,22,23,24,25,26,27,28]. ADT is recommended by international guidelines to be continued as a basic treatment for a lifelong period.

Improvements in treatment for patients with metastatic cancer have significantly prolonged survival and in essence changed this disease to a “chronic” disease with median survival times of around 5 years. This is an important improvement for patients, but it also means that side effects from treatment, including sleep disturbances, are more likely to have a lasting impact on quality of life.

### 1.1. Definition, Diagnosis and Underlying Mechanisms of Sleep Disturbances in Cancer Patients and, in Particular, Prostate Cancer Patients

Daily functioning and quality of life are reported as influenced not only by the diagnosis of prostate cancer, but also by its treatment [29]. In quality-of-life questionnaires, such as the functional assessment of cancer therapy–prostate cancer (FACT-P), one question addresses sleep quality in the functional well-being section, recognizing that sleep disturbances represent a common issue in patients with prostate cancer [30]. Sleep disturbance is often evaluated only in relation to the urge to urinate, in quality-of-life questionnaires—for example, the EORTC QLQ-PR25 [31]. Besides the notorious detrimental effect on cognitive functions and physical prowess, sleep disturbances have a multisystem impact which does not spare cardiovascular and immune function [32,33,34,35].

The last version of the International Classification of Sleep Disorders (ICSD) subdivides all entities into six thematic chapters, each of which includes several nosographic disturbances: (1) insomnia, (2) hypersomnia, (3) sleep-related breathing disorders, (4) circadian disorders, (5) sleep-related movement disorders, (6) parasomnia (behavioral abnormalities during sleep, such as sleepwalking and terrific dreams), and a last chapter with a miscellanea [36,37]. Insomnia is characterized by a difficulty in initiating sleep, maintaining sleep continuity or having poor sleep quality, where these symptoms occur despite the presence of adequate opportunity and circumstance for sleep, ultimately resulting in daytime dysfunction [37]. Chronic insomnia is defined as occurring at least three nights a week for more than three months. Insomnia has a negative impact on daytime cognition, mood, daily functioning and fatigue [38]. Insomnia can further contribute to the development of depression and difficulties in daily life, employment and relationships [39]. Between 25% and 40% of prostate cancer patients in studies report having poor sleep quality [40], and insomnia is a common disorder in the general cancer population and typically affects cancer patients after interventions (surgery, radiotherapy) or systemic treatment such as chemotherapy or hormonal therapy, which last for several months [41]. Insomnia is also the condition most frequently reported by this patient population [15].

The mechanisms underlying sleep disorders in patients with cancer are not yet fully understood. The literature suggests that sleep disorders could be related to many factors. As early as 1991, Spielman had explained the onset of insomnia in cancer patients through the theory of the three Ps [42]: predisposing factors, precipitating medical factors and perpetuating factors. The predisposing factors are age, sex, genetic factors and family predisposition. The precipitating medical factors include type of tumor, drugs administered and their side effects, such as nausea, pain, hot flashes etc., but also the type of treatment for cancer. The perpetuating factors are emotional components, such as anxiety and depression, and unhealthy sleep hygiene behaviors, such as alcohol or caffeine consumption and unhealthy nutrition [43].

Several studies have investigated the mechanisms underlying sleep disturbances and poor sleep quality in patients with breast, head and neck and endometrial cancer, but there are very few data about sleep disorders and their underlying mechanisms in patients with prostate cancer [44,45].

### 1.2. Methodologies for Investigating Sleep Disturbances in Prostate Cancer Patients

Sleep disorders can be assessed subjectively using self-reported questionnaires such as the Insomnia Severity Index (ISI) and Pittsburgh Sleep Quality Index (PSQI). Recently, actigraphy and polysomnography (PSG) have become fundamental in providing objective measures of sleep quality and for the detection and diagnosis of common sleep disturbances. Total sleep time (TST), sleep efficiency (SE), wake after sleep onset (WASO), sleep onset latency (SOL) and the arousal index (AI), are conventional instrumental parameters for assessing sleep quality employed in daily clinical practice. Indeed, PSG is the gold standard for measuring sleep, wake time, sleep stages and detecting primary and secondary sleep disorders.

Actigraphy is a portable accelerometer device worn on the non-dominant wrist, which provides an objective picture of the wearer’s sleep–wake cycle, for the duration it is worn. One or more weeks of wearing has an accuracy of around 85% with respect to PSG, although it provides no information on the architecture of sleep stages [46]. As compared with PSG, actigraphy is known to overestimate sleep and underestimate wake time, with lower levels of agreement for sleep measures that depend upon correct identification of being awake, such as sleep onset latency, sleep efficiency and wake after sleep onset. These measures may be biased by increased or decreased motor activity, the actigraphy technology, software or settings used. Considering the usual older age profile of patients with prostate cancer, it is important to point out that consistency between actigraphy and PSG measures is weaker in patients aged 60 years or older, compared with younger adults [47].

### 1.3. Rationale and Objectives of the Systematic Review

As demonstrated for other types of cancer, sleep disturbances in PCa patients are very frequent and have a potential impact on quality of life. This evidence justifies the need for a more accurate and comprehensive investigation. The current literature on the field presents methodological shortcomings, exhibiting scanty and divergent results. At present, no systematic review has been published focusing on sleep problems in PCa patients, though existing, for example, for breast cancer [48]; thus, this issue is becoming increasingly important because of the longer duration of treatment and the improved patients’ survival. This systematic review aims to assimilate current evidence on sleep disturbances in PCa patients receiving active treatments, with the objective of stimulating future research to focus on and better understand the causative relationships between the local and systemic treatments for prostate cancer and sleep quality changes.

## 2. Materials and Methods

### 2.1. Literature Search Method and Evidence Acquisition

Search Strategy: We used the PRISMA (Preferred Reporting Items for Systematic Reviews and Meta-Analyses) guidelines as a model for conducting this review [49] (Appendix A). A search of the scientific literature was conducted on PubMed, Web of Science and Cochrane Library up to the date of 21 July 2021. Searches were limited to human studies and English language articles from 1 January 1990 onwards.

We used the PICO frame (patients (P)/intervention (I)/comparison (C)/outcome (O)) for our search strategy: P = prostate cancer patients/I = treatment for prostate cancer/C = patients without treatment/O = sleep quality. The following search terms were used, combining (AND/OR) the essential terms prostate cancer and sleep: sleep disturbances, sleep disorders, insomnia, sleep quality, sleep wake disorders, sleep deprivation, androgen deprivation therapy, androgen-receptor targeted agents, novel hormonal agents, radiotherapy, prostatectomy and chemotherapy.

Our inclusion criteria required that a study attempted to quantitatively measure sleep quality and included patients in active treatment, with no history of other cancers besides prostate cancer, in either a prospective longitudinal or cross-sectional design. Studies with an interventional design, risk assessments for developing prostate cancer and studies investigating populations with multiple different tumors were excluded.

Two reviewers carried out independently the literature search using the selected database and search strategy, the data extraction and the quality assessment of the included studies. Disagreements were solved with a consensus meeting among the reviewers.

### 2.2. Quality Assessment

Quality assessment of the included studies was performed by using the quality assessment tool of the National Heart, Lung and Blood Institute (NHLBI) for observational and cross-sectional studies: https://www.nhlbi.nih.gov/health-topics/study-quality-assessment-tools, accessed on 31 January 2022. The outcomes of quality assessment are reported in Table 1.

## 3. Results

### 3.1. Results of the Literature Search

The detailed plan of the structured literature search and selection process is outlined in the PRISMA flow diagram (Figure 1). In total, 904 records were identified through the database search, after entering the initial search terms and removing duplicates, and forty-five full-text articles were assessed for eligibility, of which 16 studies met the required literature search criteria. We performed a narrative synthesis of the diagnosis and mechanisms underlying sleep disorders and the methodologies for their investigation. We then reviewed sleep disorders in relation to specific types of treatment for prostate cancer.

#### Characteristics of the Studies Included

The 16 studies recruited a total of 1271 prostate cancer patients undergoing active treatment in prospective longitudinal studies and 4209 patients in two studies with a cross-sectional design (Figure 1). They were mostly conducted in North America (*n* = 11) and the median year of publication was 2012 (range 2005–2021). No randomized controlled trials were identified. Investigating sleep quality in patients with prostate cancer has so far mostly been conducted using subjective questionnaires alone. Eleven out of sixteen studies included in this review evaluated sleep quality without an objective measure. Actigraphic measurements were used in five trials including PCa patients under active treatment, of which three focused on patients receiving ADT therapy. The first study recruited 60 patients with prostate cancer undergoing ADT, without a control group, and performed actigraphy at a single time point [52]. In another study involving 78 ADT recipients, 99 prostate cancer patients not treated with ADT and 108 healthy subjects, actigraphy was performed without baseline measurement and by means of one time point, a three-day recording [51]. The most recent study (21 July 2021), which enrolled only 24 subjects, included a 7-day actigraphic assessment before and 12 months after beginning ADT [62]. Neither in-lab nor home PSG measurements have been used as a methodology to prospectively investigate sleep quality in patients with prostate cancer. Results are graphically summarized in Figure 2.

### 3.2. Evidence Synthesis of Prostate Cancer Treatments and Sleep Disorders

#### 3.2.1. Androgen Deprivation Therapy (ADT)

In our systematic review, eight studies investigated the association between ADT and sleep disorders in a pre-defined cohort, and three more studies included patients that had received ADT at some earlier point or as an ongoing treatment during sleep assessment. In one of the studies, a prospective longitudinal study, Savard et al. [50] evaluated the evolution of insomnia following the initiation of adjuvant ADT in patients with prostate cancer. Participants’ self-report scales, including the Insomnia Severity Index and the Physical Symptoms Questionnaire, were used. Overall, participants reported significantly higher ISI scores when exposed to hormonal therapy, at each investigated time point. Somatic symptoms seemed to play a major role in the genesis of insomnia in ADT-treated patients. In particular, the acute effect was mediated by night sweats (48.6% of the total effect), nocturnal dyspnea (18.0% of the total effect), urinary symptoms (16.6% of the total effect) and pain (13.8% of the total effect), whilst the effect at the later time point was mediated only by night sweats (45.8% of the total effect). The increase in ISI score was transient and lasted around 6 months, possibly suggesting the development of an adaptation to ADT side effects.

An observational, prospective, cohort study by Brian D. Gonzalez and colleagues [51] recruited 78 ADT recipients and 207 control subjects. Patients completed the ISI questionnaire and underwent a 3-day actigraphic recording at the 6-month assessment point. ADT recipients reported worse sleep quality, higher rates of clinically significant sleep disturbances, greater hot flash interference and more severe nocturia than controls. Furthermore, they described that nocturia and hot flashes, respectively, mediated the association between objective and subjective measurements of sleep disturbances in ADT patients. As would be expected, nocturia was correlated to worse wake after sleep onset (WASO) with higher hypnic fragmentation in the actigraphic recordings.

In an observational, cross-sectional study by Hanisch et al. [64], 60 patients undergoing ADT were investigated using actigraphy, daily diaries, the Epworth Sleepiness Scale (ESS) and the general version of the Functional Assessment of Cancer Therapy (FACT-G). In particular, ADT recipients showed a sleep latency (SL) longer than 30 min, and a total sleep time of 5.9 h. Nocturia was the most frequent cause of night-time awakenings (a mean of 1.75 times per night) followed by hot flashes (a mean of 0.54 times per night).

Koskderelioglu et al. [53] enrolled 106 patients with prostate cancer, of whom 48 received ADT. Compared with the non-ADT group, patients receiving ADT showed higher levels of depression, worse sleep quality, assessed by the ISI and PSQI, and more severe fatigue. They found no significant difference among the two groups regarding excessive daytime sleepiness. In contrast, a study evaluating psychological distress in men with prostate cancer receiving ADT showed no significant difference in total PSQI score for ADT patients compared to subjects not receiving ADT, although there was a difference in daytime dysfunction [54].

Another longitudinal study involving 250 patients undergoing ADT investigated the presence of vasomotor symptoms by means of specifically designed questionnaires [55]. Around 80% of ADT patients showed varying degrees of toxicity (41.6% mild, 26.8% moderate, 11.6% severe symptoms), with sleep disturbances, hot flashes and night sweats as the most frequently reported side effects. Adverse effects were positively correlated with BMI and negatively correlated with age, but not with the duration of hormonal treatment.

The study of Tulk et al. [62] examined whether fluctuations in sleep quality and other physical symptoms are associated with changes in cancer-related cognitive impairment. They also performed actigraphy of 7 days at baseline and after 12 months of ADT. They found that fatigue and subjectively estimated wake after sleep onset (sleep diary), but not actigraphic parameters, were predictors of subjective cognitive decline in the first 12 months of ADT. Objective and subjective sleep quality worsened in patients with cognitive decline after ADT and slightly ameliorated in the others.

Finally, Sánchez-Martínez et al. demonstrated in 33 subjects worse sleep quality, as measured by Athens Insomnia Scale (AIS), after 1 year of follow-up (first evaluation within six months to one year of ADT) [63]. All these studies suggest that ADT may play a fundamental role in the onset of sleep disorders in patients with prostate cancer. Although the underlying mechanisms are not well understood, it seems that, above all, the vasomotor side effects such as hot flashes and night sweats have a significant impact on prostate cancer patients’ sleep quality.

#### 3.2.2. Radiotherapy for Localized PCa (Primary Curative or Adjuvant)

We found four trials investigating the influence and correlation of curative primary or adjuvant radiotherapy in prostate cancer, on the development of sleep disorders. Most radiotherapy only trials also included a second tumor cohort, with breast cancer patients.

Miaskowski et al. [56] evaluated sleep quality in 82 patients with early-stage cancer and no distant metastases or recurrent disease during and after radiotherapy treatment. This was a prospective, single-cohort study. Self-reported sleep disturbances (General Sleep Disturbance Scale) increased during RT (weeks 1 to 9) and then declined after the completion of RT, reaching the lowest value in the third month after the end of the treatment. Moreover, patients exhibited different degrees of insomnia in relation to the presence of inter-individual variants such as anxiety, depression and age. Indeed, patients with higher levels of anxiety and depression reported higher levels of sleep disturbances during RT. Younger age was identified as an important predictor of sleep disturbances.

In a small study, Thomas and colleagues [65] also noted an improvement in sleep quality after the end of radiotherapy. They evaluated 56 patients, of whom 23 had prostate cancer, at eight time points before, during and after treatment, using the Medical Outcomes Study—Sleep Scale (MOS-Sleep). Patients with PCa showed poorer sleep quality at baseline and in the first two weeks of treatment compared to the normative data of MOS-Sleep, with a progressive improvement over time.

Kristin Garrett et al. [58] compared sleep disturbances in 78 patients with breast cancer and 82 with prostate cancer undergoing RT as part of the primary treatment plan. Although patients with BC showed greater self-reported sleep disturbances, when evaluated with actigraphy, PCa was associated with a higher percentage of wake after sleep onset, lower total sleep time and worse sleep efficiency compared to BC patients.

Holliday et al. performed actigraphy in 28 men with early-stage (T1–T2) prostate cancer treated with RT. The researchers found no significant correlation between RT and sleep disturbances, even reporting an improvement after RT, compared to baseline, regarding sleep latency and sleep efficiency. There were two subjects with exceptionally high sleep latencies, which may have influenced the results [59]. When these two outliers were excluded, the results did not remain significant (*p* = 0.20).

To conclude, all but one study did show a negative effect of RT on sleep quality. The studies were small and almost all of them assessed sleep using subjective measures, while actigraphy was used only twice. The mechanism by which radiotherapy may induce sleep disturbances in patients with prostate cancer has not been investigated in these studies, though it was hypothesized that it might depend on the early or advanced stage of the oncological disease [59].

#### 3.2.3. ADT Combined with Radiotherapy (Primary Curative or Adjuvant)

The combination of RT and ADT is recommended for prostate cancer with intermediate–unfavorable and high-risk disease, according to international guidelines. Their combined use in these patient groups shows an increase in disease-free and progression-free survival [66,67,68,69,70,71,72]. Only one study addressing sleep disturbances related to this treatment combination was detected in our systematic literature review.

Savard et al. [40] found an additive effect of ADT on the negative impact on sleep already documented for RT. Specifically, they found a higher ISI score in the ADT plus RT patients group (*n* = 28) than in the RT alone group (*n* = 32). Insomnia, classified by an ISI score > 8 points, was reported in 22% of patients in the ADT plus RT group vs. 14.3% in the RT alone group. Moreover, while sleep quality remained stable in the RT group during treatment, the percentage of patients being affected by sleep disturbances increased from 22% at baseline to 41.9% in ADT plus RT patients at 6 months. No main effects (group, time or interaction) were found to be significant, in part because of the lower sensitivity (power level) of the categorical analysis (generalized mixed model). The ADT plus RT group was divided into two subgroups, long-term ADT and short-term ADT. In both subgroups, there was an increase in the ISI score, which remained stable for the duration of the treatment. They also showed that both hot flashes/night sweats and urinary symptoms were correlated with higher ISI scores. This correlation was more significant in the ADT plus RT patient group compared to the RT group. This single study underlines that RT combined with ADT is associated with an increase in insomnia severity. Of note, this seems not to be a direct mechanistic treatment effect, but is rather mediated by somatic symptoms such as hot flashes and urinary disorders. However, the ISI scale is a measure of symptom severity and does not provide a diagnosis of insomnia, which instead requires the satisfaction of specific diagnostic criteria. Further data are necessary to explain the possible synergistic mechanism behind the negative impact of the combined treatment of radiotherapy and ADT on sleep quality.

#### 3.2.4. Prostatectomy

Radical prostatectomy represents a standard treatment in patients with mostly intermediate risk and in selected cases also for high-risk or locally advanced patients. We found only one article focusing on the effect of prostatectomy on sleep quality to be included in this review.

The research group of Savard J and colleagues [15] evaluated the prevalence and risk factors for insomnia in 327 prostate cancer patients treated with radical prostatectomy alone. Sleep disturbances were assessed using a battery of questionnaires on sleep and related issues (for example, anxiety, depression, fatigue, quality of life). This prospective study showed that 31.5% of patients who underwent prostatectomy reported non-specific sleep disturbances; 18% suffered from insomnia, of whom 95% had chronic insomnia. Insomnia was diagnosed after prostate cancer diagnosis in 50% of cases. In addition, young age, unfavorable prognosis, depression, anxiety, abdominal pain and climacteric symptoms were all considered risk factors for the development of poor sleep quality.

#### 3.2.5. Novel Hormonal Agents

The literature is scarce on the effects of novel hormonal agents on sleep quality, except only one letter to the editor that was not included in this systematic review. The authors reported the case of a 58-year-old man who developed sleep apnea 6 weeks after starting treatment with enzalutamide [73]. Enzalutamide is generally well tolerated, and studies reported a favorable toxicity profile. Some of these side effects have been shown to mediate lowered sleep quality. Moreover, fatigue and asthenia could be a sequela of sleep disturbance, not being adequately assessed and therefore often underreported. A possible correlation between enzalutamide and other novel hormonal agents such as Apalutamide and Darolutamide and sleep disorders can therefore not be ruled out, although further studies including adequate methodology for investigating sleep quality are needed to further support this hypothesis. In all large phase III trials investigating novel hormonal agents plus ADT, compared to placebo or standard of care (SOC), sleep disorders have not been reported or assessed as independent adverse events (Table 2), as is also not the case in trials of the CYP17 inhibitor Abiraterone (Table 3).

## 4. Discussion

This is the first systematic review to synthesize the literature focused on changes in sleep quality and the development of sleep disturbances in patients treated with different modalities for prostate cancer. Prostate cancer diagnosis is frequently associated with sleep-related abnormalities, resulting in worsened quality of life for patients [1,83,84]. Between 20% and 25% of PCa patients regularly use pharmacological remedies to improve sleep [15,85]. The literature suggests a multifactorial etiology for sleep disruption in patients undergoing PCa treatments: type of treatment, side effects and individual patient factors [43,86].

Prostate cancer treatments can probably unbalance the level of inflammatory cytokines involved in sleep-related molecular patterns [87,88]. Interleukin-1 (IL-1) and tumor necrosis factor alpha (TNF-alpha), for example, promote non-rapid eye movement (NREM) sleep, while a low IL-6 concentration is thought to worsen sleep quality indicators. Other cytokines deemed to be involved in sleep disturbances are IL-4, IL-10 and transforming growth factor beta (TGF-β), which may have sleep-inhibiting properties by mechanisms that are not yet fully understood [89].

We have noted in the trials selected in this systemic review that all the available treatments for PCa are accompanied by some sort of sleep disruption, varying in quality and intensity. Anti-hormonal therapy by androgen deprivation (castration) may cause insomnia or sleep disruption via two mechanisms: primarily, by alterations of the levels of hypothalamic hormones involved in sleep regulation [90]; secondly, through an indirect mechanism mediated by climacteric symptoms [51]. Hot flashes are commonly present in patients treated with ADT, reported in around 80% of patients, with 27% of the patients considering it as the most impactful adverse effect. There are no validated treatments for this side effect, and experts use different approaches [91]. Patients treated with radiotherapy with or without ADT report nocturia, defined as the need to urinate more than twice a night, as the most common cause of sleep disorders [51,56,92]. Sleep disturbances increased during radiotherapy and then declined after the completion of the treatment. The inflammatory cascade activated by irradiation may have a relevant role in sleep disruption. 

Prostatectomy and radiotherapy, curative treatments in early-stage patients, have been associated with sleep disturbances and poor sleep quality in most of the reviewed studies. Otherwise, studies investigating sleep changes in metastatic PCa during systemic chemotherapy, ADT and novel endocrine agents are lacking.

One single study, although limited by a small sample size (24 subjects), hypothesized a role of sleep disturbance in the appearance of cancer-related cognitive decline after ADT [62].

Finally, a correlation between sleep disturbance and anxiety and depression [53] has been reported. There are data suggesting that men receiving ADT are at higher risk of developing major depression after five years, compared to men not receiving ADT. Therefore, it is not clear if ADT is an independent driver for the development of sleep disturbances or if other long-term side effects caused by ADT, such as depression, indirectly induce longer-term sleep problems [93].

Most of the studies included in our systemic review demonstrated an association between prostate cancer treatments and sleep disturbances; however, they often have limitations. The majority of the studies have a transversal or cross-sectional design. Therefore, interpretation of the results must be done critically, especially regarding causal interpretations. Many of the studies feature a small sample size, different prostate cancer stages or include patients with other previous treatments, which could have been confounding factors in the results.

Self-report measures of sleep quality are by far the most used, and objective measurements (in the form of actigraphy) are provided only by a few studies, with some methodological limitations potentially bringing results into question.

Video-polysomnography (PSG) is the current gold standard for measuring sleep, providing complete information regarding sleep macro- and microstructure, sleep-related EEG features and the occurrence of other sleep-related disorders in addition to insomnia [94,95,96,97]. Several sleep-related abnormalities, such as central or obstructive breathing disorders, periodic limb movements, parasomnia and morpheic epilepsy, are not detected, except with a whole-night video-PSG recording across its full spectrum of measurements [98]. Moreover, PSG is a flexible tool, which can be adapted in its montage with different combinations of sensors, in order to detect those biological parameters considered informative for the sleep disorder that is suspected. The PSG identifies and quantifies (scoring) a large spectrum of sleep disorders, with high accuracy in particular on both central and obstructive sleep-related breathing disturbances, which may occur in oncological diseases involving the head and neck [99,100,101]. In 2008, a study by Kathy P. Parker et al. found a reduction in SWS and REM sleep in patients with different types of advanced cancer [102]. This result was possible only through the PSG. Another polysomnographic study in breast cancer patients permitted researchers to identify PLM as the only variable discriminating subjects with or without insomnia [103]. Notably, no study assessed sleep quality changes during PCa treatment using polysomnographic methods. The importance of patient-reported outcomes is currently well recognized in clinical research and also in clinical practice. However, subjective reported measurements of sleep disorders are limited, and sometimes not very reliable. Sleep misperception, which is characterized by a marked mismatch between subjective and objective measures, is very common in patients with insomnia, especially in elderly subjects, with a tendency to underestimate their sleep quality. There is no a priori reason to think that sleep misperception would not occur in patients with cancer, so subjective measurements should be verified with concordant objective measurements. It would therefore be wise to define sleep abnormalities in patients with PCa using prospective, controlled, combined subjective and objective investigations.

### 4.1. Limitations and Strengths

Our method of critically selecting only full-text articles focusing on a high number or percentage of prostate cancer patients treated allowed us to underline the lack of research in this field, especially lacking important publications in the last two years. Larger randomized trials are missing and our systematic review stresses the need of evaluating sleep disorders as a side effect, also in the recently authorized systemic treatments in metastatic PCa. The review was performed in accordance with a systematic review expert at our institution. Limitations include the nature of all included studies not being randomized trials and most of them retrospective and even some of them cross-sectional. The general quality of studies in this field is limited also due to the heterogeneity of methodologies used and reported in these studies.

### 4.2. Suggestions for Future Studies

To date, there is great uncertainty about the true prevalence and severity of sleep disorders in PCa patients under different treatments, especially with ADT and novel hormonal agents in the metastatic setting. Further studies should focus on implementing a combination of subjective and objective measures, high-quality methodologies and instruments for assessing sleep disorders at baseline and regular pre-defined timepoints. This would help to better understand the prevalence, severity and timing of sleep disorder appearances in PCa patients under active treatments. Results of these trials could help to focus on assessing the side effects and, ultimately, to plan interventions. Moreover, randomized trials investigating local and systemic treatments in PCa should focus on implementing questionnaires and the reporting of sleep disorder symptoms and insomnia as adverse events.

## 5. Conclusions

The results of our systematic review confirm that in the majority of our selected studies, PCa treatments seem to increase the risk of patients developing sleep disorders. These often seem to be mediated by treatment side effects, such as hot flushes and nocturia. However, the methodologies used in the published trials prevent a rigorous mediation analysis. Future studies are warranted to confirm the high prevalence of sleep disorders in PCa patients, to quantify the severity of these sleep disorders, to use accurate diagnostic procedures for sleep disturbances, such as polysomnography, and to measure their impact on oncological outcomes. A structured plan of research in the field would pave the way towards understanding how PCa treatments affect changes in sleep quality.

This improved knowledge of the pathogenesis of sleep disorders in these patients will then facilitate the development of pharmaceutical and non-pharmaceutical strategies aimed at mitigating treatment-related sleep disturbances, thereby improving sleep quality, psychological health and quality of life in men with PCa.

## Figures and Tables

**Figure 1 cancers-14-01784-f001:**
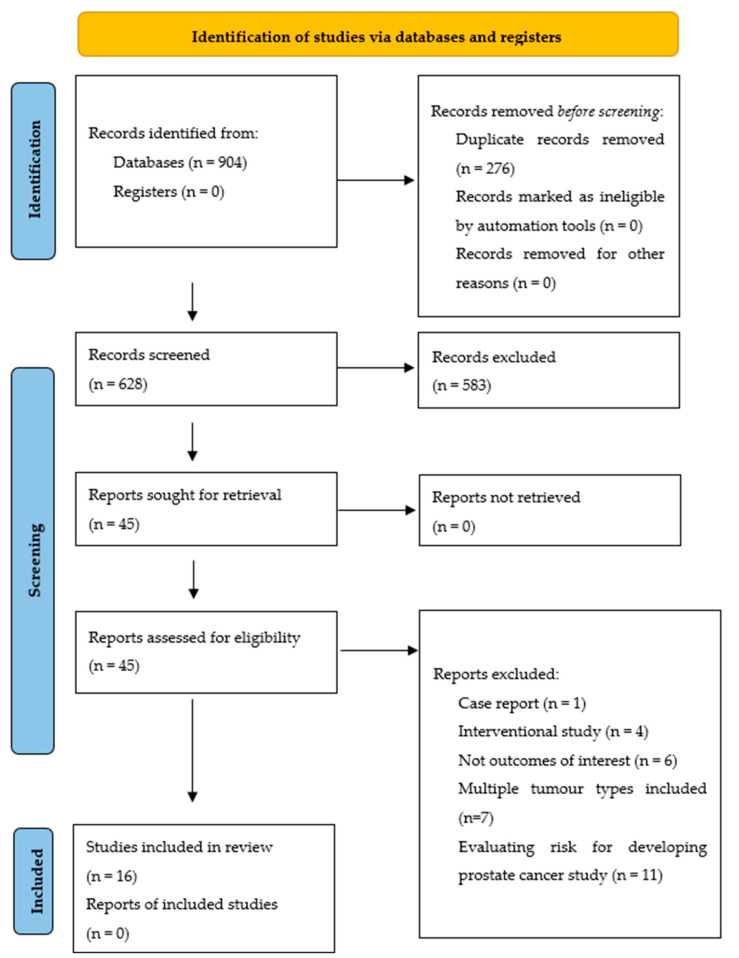
PRISMA flow diagram.

**Figure 2 cancers-14-01784-f002:**
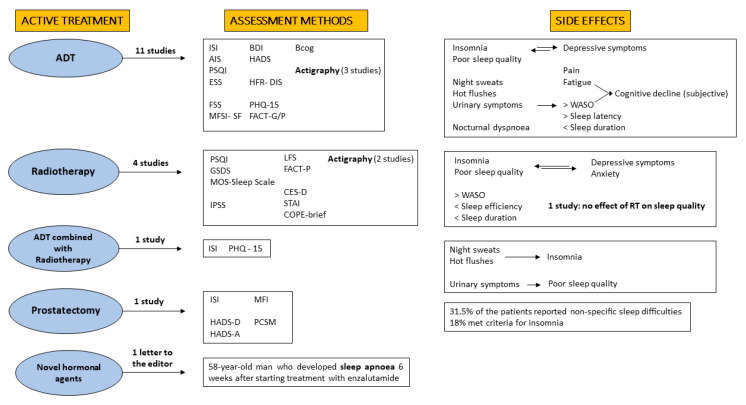
Sleep disturbances in prostate cancer active treatments: graphical summary of the literature synthesis. Androgen deprivation therapy (ADT), Insomnia Severity Index (ISI), Athens Insomnia Scale (AIS), Pittsburgh Sleep Quality Index (PSQI), Epworth Sleepiness Scale (ESS), Fatigue Severity Scale (FSS), Multidimensional Fatigue Symptom Inventory—Short-Form (MFSI-SF), Beck’s Depression Inventory (BDI), Hospital Anxiety and Depression Scale (HADS), Hot Flash Related Daily Interference Scale (HFR-DIS), Physical Symptoms Questionnaire (PHQ-15), Functional Assessment of Cancer Therapy—General (FACT-G), Functional Assessment of Cancer Therapy—Prostate (FACT-P), Brief Scale for Cognitive Evaluation (BCog), General Sleep Disturbances Scale (GSDS), Medical Outcomes Survey—Sleep Scale (MOS-Sleep Scale), International Prostate Symptom Score (IPSS), Lee Fatigue Scale (LFS), Center for Epidemiological Studies—Depression Scale (CESD), Spielberg State-Trait Anxiety Inventories (STAI), Coping Orientation to Problems Experienced Inventory—brief (COPE-brief), Hospital Anxiety and Depression Scale (HADS-D), Hospital Anxiety and Anxiety Scale (HADS-A), Multidimensional Fatigue Inventory (MFI), Prostate Cancer-Specific Module (PCSM), Wake After Sleep Onset (WASO), Radiotherapy (RT).

**Table 1 cancers-14-01784-t001:** Studies reporting on prostate cancer treatments and sleep disorders using subjective and objective measurements.

Authors	Study Design	Number of Participants	Inclusion/Exclusion Criteria	Methods	Results	Quality Assessment
Savard, J. et al., 2015 [50]	Prospective, cohort	Tot. *n* = 728 BC *n* = 465 PC *n* = 263 treatment during study (RT 4.2%, CHT 0.8%, hormone therapy 0.4% (76% LHRH, 76% bicalutamide))	Inclusion PCa: non-metastatic, after prostatectomy Exclusion: neoadjuvant cancer treatment; brachytherapy, severe cognitive impairment or severe psychiatric disorder; sleep disorder	ISI PHQ-15	PCa was consistently associated with insomnia, and this association was strongly mediated by night sweats. Significantly higher ISI scores at 14, 16 and 18 months when exposed to hormone therapy.	good
* Gonzalez, B.D. et al., 2018 [51]	Prospective cohort study	Tot. PCa 177 PC *n* = 99 (receiving ADT *n* = 78) Control group no cancer *n* = 108	Inclusion: ADT for non-metastatic or asymptomatic metastatic PCa, ADT for ≥6 months. Patients not treated with ADT with non-metastatic prostate cancer treated only by prostatectomy for prostate cancer, and not receiving testosterone supplementation	ISI HFR-DIS Actigraphy (3 days at one time point at 6 months)	ADT recipients reported worse subjective sleep disturbances over time. Nocturia mediated the association between ADT and objective sleep disturbances. Hot flash interference mediated the association between ADT and subjective sleep disturbances.	good
* Hanisch, L.J. et al., 2011 [52]	Cross-sectional	*n* = 60 ADT only	Inclusion: ongoing ADT, exclusion: recent surgery, radiation, chemotherapy or myelosuppressive medication 49% metastatic, 30% BCR, 21% localized disease at time of enrolment	Actigraphy (7 days, one time point) Daily Diary ESS FACT-G	ADT associated with sleep disturbances. Patients receiving ADT had lower sleep quality with difficulty in falling asleep, sleep fragmentation and daily napping. They presented a reduced TST (6 h), but no interference with the activities of daily life. Nocturia and hot flashes were common causes of sleep disruption.	good
Koskderelioglu, A. et al., 2017 [53]	Cross-sectional	Tot. 106 prostatectomy adj. ADT > 6 months *n* = 48 no adj. ADT *n* = 58	Inclusion: prostatectomy Adj. ADT or follow-up only Exclusion: patients with major stroke, sleep disorders, dementia, Parkinson’s disease, traumatic brain injury, epilepsy and psychiatric condition	PSQI BDI ESS FSS	ADT patients reported higher levels of depression, worse quality of sleep and more severe fatigue (*p* < 0.001). PSQI scores showed a positive correlation with BDI and FSS scores. ADT was strongly associated with PSQI and FSS at multivariate analysis.	good
Saini, A. et al., 2013 [54]	Cross-sectional	Tot. 103 ADT *n* = 49 no ADT *n* = 54	Inclusion: prostatectomy or 3D-RT, no metastatic disease; absence of major comorbidities; PS 0–1, testosterone < 0.5 ng/mL. Exclusion: history of neuropsychiatric disease or drugs, progressive disease at the study entry	FACT-P HADS BIS PSQI	No difference was found between the 2 groups for total PSQI and the other relevant items, except for daytime dysfunction (*p* = 0.03).	good
Challapalli, A. et al., 2018 [55]	Prospective, single-cohort	Tot. 250 (54% > 6 months ADT, 89% LHRH-agonists)	Inclusion: prostate cancer patients on ADT	specific questionnaire on vasomotor symptoms	80% of ADT-treated patients had sleep problems, which were more prevalent in younger patients with higher BMI.	good
Miaskowski, C. et al., 2011 [56]	Prospective, single-cohort	Tot. *n* = 82 RT (primary or adj)	Inclusion: primary or adjuvant RT, KPS > 60 Exclusion: metastatic disease, had more than one cancer diagnosis or had a diagnosed sleep disorder	PSQI GSDS CES-D STAI NRS LFS	Sleep disturbances increased during RT and decreased after the completion of RT. Younger men with co-occurring depression and anxiety had the greatest risk for sleep disturbances during RT. ADT before RT (51% of patients) and fatigue are not predictors of sleep disturbances.	good
Thomas, K.S. et al., 2011 [57]	Prospective, cohort	Tot. *n* = 56 BC *n* = 33 PC *n* = 23 (primary RT)	Inclusion criteria PCa: radiation therapy for early stage Exclusion: recurrent cancer; prior or planned treatment with chemotherapy; immunosuppressive medication or tobacco.	MOS-Sleep Scale COPE-brief FACT-P	PCa: RT was associated with a decrease in TST. Sleep latency increased at the beginning of RT and during treatment, but decreased at follow-up. There was no significant change in sleep quality over the course of treatment.	fair
* Garrett, K. et al., 2011 [58]	Cross-sectional	Tot. 160 BC *n* = 78 PC *n*= 82 (RT primary or adj.)	Inclusion criteria PCa: primary or adjuvant RT; KPS > 60 exclusion: metastatic disease; more than one cancer diagnosis; sleep disorder	PSQI GSDS LFS Actigraphy (48 h one time point)	Results PCa: Sleep disturbances and fatigue are significant burdens. Significantly lower TST, lower sleep efficiency and higher percentage of WASO compared to patients with BC.	good
* Holliday, E.B. et al., 2016 [59]	Prospective, single-cohort	Tot. 28 all RT	Inclusion: EBRT for T1-T2 PCa. Exclusion criteria: concurrent ADT; brachytherapy; psychiatric disorders treatment for any cancer	IPSS Actigraphy	Sleep efficiency improved during radiotherapy, fatigue increased and was associated with reduced QoL.	good
Savard, J. et al., 2013 [40]	Prospective, cohort	Tot. 60 RT + ADT *n* = 28 RT *n* = 32	Inclusion: non-metastatic prostate cancer, scheduled to receive curative RT only or RT plus ADT; Exclusion: prior history of cancer; score <24 on the Mini-Mental State Examination, any treatment for cancer	ISI PHQ	A significant interaction effect was found indicating an increase in insomnia scores in ADT + RT patients at 2, 4 and 6 months, as compared with baseline, and stable scores in RT only patients. A significant mediating role of hot flashes and night sweats was found in the relationship between ADT and insomnia, while nocturia mediated the association between RT and poor sleep quality.	good
Savard, J. et al., 2005 [15]	Cross-sectional	Tot. 327 all RP	Inclusion: radical prostatectomy for prostate cancer within the past 10 years.	ISI HADS-D HADS-A MFI PCSM	31.5% of the patients reported non-specific sleep difficulties and 18% of them met criteria for insomnia. In 95% of the cases, insomnia was chronic. In 50% of patients with insomnia, the onset of sleep difficulties followed the cancer diagnosis. Risk factors for insomnia were younger age, worse prognosis, intestinal pain, depression and ADT-related symptoms (for patients undergoing ADT).	good
Maguire, R. et al., 2018 [60]	Cross-sectional design	*n* = 3348	Inclusion: being at least 2 years post diagnosis	EORTC QLQC30 QLQPR25 EQ5D-5L	Sleep disturbances have a positive association with side effects such as urinary symptoms, hormone treatment-related symptoms, intestinal symptoms and depression/anxiety.	good
Hervouet, S. et al., 2005 [61]	Cross-sectional	Tot. 861 RT *n* = 392 BR *n* = 188 RP *n* = 28 Current hormone therapy (10.2%; 4.8%; 20.6%) Lifetime hormone therapy (93.6%; 77.1%; 54.5%)	Inclusion: RT, BR, RP as an initial treatment for PC within the past 7 years; age < 80 at study entry Exclusion: any other type of cancer; orchiectomy; chemotherapy; severe cognitive impairment	HADS-D HADS-A ISI MFI PCSM EORTC QLQC30	Sexual difficulties were the most frequently reported (70.5%), followed by insomnia (31.9%), anxiety (23.7%), fatigue (18.5%) and depression (17.0%). Patients treated with RT had higher levels of clinically significant insomnia (*n* = 137; 35%) compared to men receiving RP (*n* = 84; 30%), scores of fatigue motivation were higher in ongoing hormone therapy group.	good
* Tulk, J. et al., 2021 [62]	Prospective, single-cohort	*n* = 24	Inclusion: ADT after RT; age > 18 at study entry Exclusion: prior history of cancer diagnosis and treatment.	FACT-Cog ISI PSQI Sleep Diary HADS MFSI-SF Actigraphy HFR-DIS	The worsening of subjectively estimated wake after sleep onset (sleep diary) was a predictor of subjective cognitive decline in the first 12 months of ADT.	good
Sánchez-Martínez, V. et al., 2021 [63]	Prospective, single-cohort	*n* = 33	Inclusion: ADT with or without previous prostatectomy Exclusion: history of other chemotherapy treatment for prostate or any other cancer, cognitive deterioration, relevant change in the health status that could influence sleep quality, mood or cognitive performance.	AIS BCog GDS	Lower subjective sleep quality and more depressive symptoms after one year of follow-up (first assessment in the six months to one year treatment with ADT).	fair

Pittsburgh Sleep Quality Index (PSQI), Physical Symptoms Questionnaire (PHQ-15), General Sleep Disturbances Scale (GSDS), Lee Fatigue Scale (LFS), Insomnia Severity Index (ISI), Hot Flash Related Daily Interference Scale (HFR- DIS), AIS (Athens Insomnia Scale), International Prostate Symptom Score (IPSS), Center for Epidemiological Studies-Depression Scale (CESD), Spielberg State-Trait Anxiety Inventories (STAI-S and STAI-T), Numeric rating scale (NRS), Lee Fatigue Scale (LFS), Hot Flash Related Daily Interference Scale (HFR- DIS), Epworth Sleepiness Scale (ESS), Fatigue Severity Scale (FSS), Beck’s Depression Inventory (BDI), Functional Assessment of Cancer Therapy—General (FACT-G), European Organisation for Research and Treatment of Cancer Quality of Life Questionnaire (EORTC QLQC30), Quality of Life Questionnaire—Prostate 25 (QLQPR25), Generic health-related quality of life (EQ5D-5L), Medical Outcomes Survey—Sleep Scale (MOS-Sleep Scale), Coping Orientation to Problems Experienced Inventory—brief (COPE-brief), Hospital Anxiety and Depression Scale (HADS-D), Hospital Anxiety and Anxiety Scale (HADS-A), Multidimensional Fatigue Inventory (MFI), Multidimensional Fatigue Symptom Inventory—Short-Form (MFSI-SF), Prostate Cancer-Specific Module Supplementing the European Organization for Research and Treatment of Cancer Quality of Life Questionnaire-C30 (PCSM), Prostate Cancer-Specific Module (PCSM), Brief Scale for Cognitive Evaluation (BCog), Functional Assessment of Cancer Therapy—Prostate (FACT-P), Body Image Scale (BIS), Breast cancer (BC), Prostate cancer (PC), Radiotherapy (RT), Brachytherapy (BR), Biochemical recurrence (BCR), Polysomnography (PSG), Magnetic resonance imaging (MRI), * = studies including objective measurements.

**Table 2 cancers-14-01784-t002:** Phase III trials including second-generation NSAA (Enzalutamide, Darolutamide, Apalutamide) reporting neurological adverse events of interest for sleep quality.

Adverse Events	PROSPER [74] *n* = 1401 nmCRPC *n* (%)		ARCHES [25] *n* = 1150 mHSPC *n* (%)		ENZAMET [26] *n* = 1125 mHSPC *n* (%)		PREVAIL [75] *n* = 1717 mCRPC Chemo Naive *n* (%)		AFFIRM [76] *n* = 1199 mCRPC after Docetaxel Failure *n* (%)		ARAMIS [77] *n* = 1509 nmCRPC *n* (%)		SPARTAN [21] *n* = 1207 nmCRPC *n* (%)		TITAN [78] *n* = 1052 mHSPC *n* (%)	
	Enza Plus ADT	Placebo Plus ADT	Enza Plus ADT	Placebo Plus ADT	Enza Plus ADT	SOC Plus ADT	Enza Plus ADT	Placebo Plus ADT	Enza Plus ADT	Placebo Plus ADT	Daro Plus ADT	Placebo Plus ADT	Apa Plus ADT	Placebo Plus ADT	Apa Plus ADT	Placebo Plus ADT
**-sleep disorder**	nr	nr	nr	nr	nr	nr	nr	nr	nr	nr	nr	nr	nr	nr	nr	nr
**-fatigue**																
all grades	303 (33)	64 (14)	138 (24)	112 (19.5)	nr	nr	310 (36)	218 (36)	260 (34)	116 (29)	115	48 (8.7)	244 (30.4)	84 (21.1)	103	86 (16.3)
G3	3 (3)	3 (1)	10 (1.7)	9 (1.6)	31 (6)	4 (1)	16 (2)	16 (2)	50 (6)	29 (7)	(12.1)	5 (0.9)	7 (0.9)	1 (0.3)	(19.7)	0
**-dizziness**																
all grades	91 (10)	20 (4)	29 (5.1)	20 (3.5)	nr	nr	nr	nr	nr	nr	4 (80.4)	22 (4.0)	75 (9.3)	25 (6.3)	8 (1.5)	nr
G3	4 (<1)	0	0	0	nr	nr	nr	nr	nr	nr	43 (4.5)	1 (0.2)	5 (0.6)	0	nr	nr
**-cognitive/memory impairment**																
all grades	48 (5)	9 (2)	26 (4.5)	12 (2.1)	nr	nr	nr	nr	nr	nr	2 (0.2)	8 (1.5)	41 (5.1)	12 (3)	nr	nr
G3	1 (<1)	0	4 (0.7)	0	nr	nr	nr	nr	nr	nr	9 (0.9)	0	0	0	nr	nr
**-syncope**																
all grades	nr	nr	nr	nr	nr	nr	nr	nr	nr	nr	0	nr	nr	nr	nr	nr
G3	nr	nr	nr	nr	20 (4)	6 (1)	nr	nr	nr	nr	nr	nr	nr	nr	nr	nr
**-delirium**																
all grades	nr	nr	nr	nr	0	1 (<1)	nr	nr	nr	nr	nr	nr	nr	nr	nr	nr
**-headache**																
all grades	85 (9)	21 (5)	nr	nr	nr	nr	91 (10)	59 (7)	93 (12)	22 (6)	nr	nr	nr	nr	nr	nr
G3	2 (<1)	0	nr	nr	nr	nr	2 (<1)	3 (<1)	6(<1)	0	nr	nr	nr	nr	nr	nr
**-seizures**																
all grades	3 (1)	0	nr	nr	nr	nr	1 (<1)	1 (<1)	5 (<1)	0	2 (0.2)	1 (0.2)	2 (0.2)	0	3 (0.6)	2 (0.4)
G3	2 (1)	0	nr	nr	nr	nr	1 (<1)	0	5 (<1)	0	0	0	0	0	0	0

NSAA = non-steroidal antiandrogen; SOC = Bicalutamide, Nilutamide or Flutamide; nr = not reported; Enza = Enzalutamide; nmCRPC = non-metastatic castration-resistant prostate cancer: mHSPC = metastatic hormone-sensitive prostate cancer; mCRCP = metastatic castration-resistant prostate cancer; Dara = Darolutamide; Apa = Apalutamide.

**Table 3 cancers-14-01784-t003:** Phase III trials with LHRH and CYP17 inhibitor (Abiraterone) reporting adverse events of interest for sleep quality.

Adverse Events	COU-AA-301 [79] *n* = 1195 mCRPC after Docetaxel Failure *n* (%)		COU-AA-302 [80] *n* = 1088 mCRPC Chemo Naive *n* (%)		LATITUDE [24] *n* = 1199 mHSPC *n* (%)		STAMPEDE [81] *n* = 1917 PC Not Previously Treated with Hormone Therapy *n* (%)		SWOG S9346 [82] *n*= 1134 mHSPC *n* (%)	
	Abiraterone Plus Prednisone Plus ADT	Placebo Plus Prednisone Plus ADT	Abiraterone Plus Prednisone Plus ADT	Placebo Plus Prednisone Plus ADT	Abiraterone Plus Prednisone Plus ADT	Double Placebo Plus ADT	Abiraterone Plus Prednisone plus ADT +/− Radiotherapy	Double Placebo Plus ADT +/− Radiotherapy	Continuous ADT	Intermittent ADT
**-sleep disorder**										
all grades	nr	nr	nr	nr	nr	nr	222 (23)	180 (19)	nr	nr
G3	nr	nr	nr	nr	nr	nr	14 (1)	6 (1)	nr	nr
**-fatigue**										
all grades	346 (44)	169 (43)	212 (39)	185 (34)	77 (13)	86 (14)	424 (45)	400 (42)	nr	nr
G3	64 (8)	36 (9)	nr	nr	10 (2)	14 (2)	15 (2)	21 (2)	nr	nr
**-fluid retention and edema**										
all grades	241 (31)	88 (22)	nr	nr	nr	nr	176 (19)	134 (14)	nr	nr
G3	16 (2)	4 (1)	nr	nr	nr	nr	5 (1)	0 (0)	nr	nr
**-back pain**										
all grades	233 (30)	129 (33)	173 (32)	173 (32)	110 (18)	123 (20)	0 (0)	0 (0)	nr	nr
G3	44 (6)	37 (9)	nr	nr	14 (2)	nr	0 (0)	0 (0)	nr	nr
**-nausea ***										
all grades	233 (30)	124 (32)	120 (22)	118 (22)	nr	nr	132 (14)	81 (8)	nr	nr
G3	12 (2)	10 (3)	nr	nr	nr	nr	1 (0)	1 (0)	nr	nr
**-arthralgia**										
all grades	215 (27)	89 (23)	154 (28)	129 (24)	nr	nr	nr	nr	nr	nr
G3	33 (4)	16 (4)	nr	nr	nr	nr	nr	nr	nr	nr
**-constipation**										
all grades	206 (13)	89 (23)	125 (23)	103 (19)	103 (19)	nr	866 (90)	660 (70)	nr	nr
G3	8 (1)	16 (4)	nr	nr	nr	nr	1 (0)	5 (1)	nr	nr
**-bone pain**										
all grades	194 (25)	110 (28)	106 (20)	103 (19)	74 (12)	88 (15)	nr	nr	nr	nr
G3	42 (5)	25 (6)	nr	nr	20 (3)	17 (3)	nr	nr	26 (3.6)	30 (4)
**-vomiting**										
all grades	168 (21)	97 (25)	nr	nr	nr	nr	63 (7)	34 (4)	nr	nr
G3	13 (2)	11 (3)	nr	nr	nr	nr	4 (0)	1 (0)	nr	nr
**-diarrhea**										
all grades	139 (18)	53 (14)	117 (22)	96 (18)	nr	nr	229 (24)	194 (20)	nr	nr
G3	5 (1)	5 (1)	nr	nr	nr	nr	13 (1)	8 (1)	nr	nr
**-muscle spasm**										
all grades	nr	nr	75 (14)	110 (20)	nr	nr	nr	nr	nr	nr
G3	nr	nr	nr	nr	nr	nr	nr	nr	1 (0)	2 (<1)
**-hot flashes**										
all grades	nr	nr	121 (22)	98 (88)	nr	nr	496 (52)	510 (53)	nr	nr
G3	nr	nr	nr	nr	nr	nr	41 (4)	39 (4)	20 (6)	16 (5)
**-spinal-cord compression**										
all grades	nr	nr	nr	nr	14 (2)	12 (2)	nr	nr	nr	nr
G3	nr	nr	nr	nr	12 (12)	7 (1)	nr	nr	nr	nr
**neurologic disorders ****										4
all grades	nr	nr	nr	nr	nr	nr	nr	nr	43 (14)	6 (14)
G3	nr	nr	nr	nr	nr	nr	nr	nr	15 (2)	15 (2)

ADT= androgen deprivation therapy; nr = not reported; mHSPC = metastatic hormone-sensitive prostate cancer; mCRCP = metastatic castration-resistant prostate cancer; PC = prostate cancer; * see nausea for nausea and vomiting adverse events; ** psychiatric disorder includes anxiety, depression, confusion, disorientation or sleep disorder.

## Data Availability

The data presented in this study are available in this article (and Appendix A).

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
