# Peer review of "Prostate Cancer and Sleep Disorders: A Systematic Review"

_cancers, 2022, doi:10.3390/cancers14071784_

Round 1

Reviewer 1 Report

The authors have done a commendable job in compiling a very pertinent review on Prostate Cancer and Sleep Disorders.Sleep disorders generally combine with cancer prognosis and treatment.The review covers well the background of sleep related problems and different stages of cancer.Different therapies like ADT,Radiotherapy, Prostatectomy,novel hormonal agents and their effects on sleep are well explained.

Author Response

We thank the reviewer for giving us such exellent feedback on our manuscript. 

Reviewer 2 Report

The systematic review by Sparasci and co-authors deals with an interesting and timely topic. The paper is scientifically sound, and well written. The authors should be complemented for including critical and insightful comments to the results from individual papers included. However, since this is a review, it is important to convey the results to the audience in a more graphical form in addition to the tables and text. I suggest that the authors include one or two new figures that summarized the main conclusions and way forward.

Author Response

We thank the reviewer for this important comment. Indeed a graphical summary of the results will help the reader to better understand the conclusion of our systematic review. We've added a Figure 2 incl title and legend in the text and as an attachment in PDF. The Figure 2 was referenced on page 15 of the manuscript: 3.2. Evidence Synthesis of Prostate Cancer Treatments and Sleep Disorders (Figure 2) 

Reviewer 3 Report

The paper deals with a systematic review regarding sleeping disorders in subjects submitted to active treatment for prostate cancer. It is well written, clear and concise. Methodology is sound. However, it is debatable to exclude studies enrolling patients on active surveillance or active monitoring or watchful waiting (if available). There is a minor spelling mistake in the abstract: "undergoing active treatment in prospective longitudinal studies and with a cross-sectional design" should be rephrased in "in either a prospective longitudinal or cross-sectional design" 

Author Response

We thank the reviewer for raising this observation. Patients on watchful waiting or active surveillance were excluded because otherwise patients' characteristics would have been too heterogenious, because active treatments in prostate cancer treatments were reported to cause sleep disorders and this was the focus of our systematic review. 

The suggested change of the sentence in the abstract has been added on page 1 line 42 of the abstract in the manuscript.